# Novel Slot Detection with an Incremental Setting

**Chen Liang[1]***, **Hongliang Li[1]***, **Changhao Guan[1]***, **Qingbin Liu[2]**,
**Jian Liu[1]†**, **Jinan Xu[1]†**, **Zhe Zhao[3]**
[1] Beijing Key Lab of Traffic Data Analysis and Mining,
Beijing Jiaotong University, Beijing, China
[2] Platform and Content Group, Tencent, China
[3] Tencent AI Lab
{21120367, 23120367, 23120344, jianliu, jaxu}@bjtu.edu.cn
{qingbinliu, nlpzhezhao}@tencent.com

## Abstract

Current dialogue systems face diverse user requests and rapid change domains, making quickly adapt to scenarios with previous unseen slot types become a major challenge. Recently, researchers have introduced novel slot detection (NSD) to discover potential new types. However, dialogue system with NSD does not bring practical improvements due to the system still cannot handle novel slots in subsequent interactions. In this paper, we define incremental novel slot detection (INSD), which separates the dialogue system to deal with novel types as two major phrases: 1) model discovers unknown slots, 2) training model to possess the capability to handle new classes. We provide an effective model to extract novel slots with set prediction strategy and propose a query-enhanced approach to overcome catastrophic forgetting during the process of INSD. We construct two INSD datasets to evaluate our method and experimental results show that our approach exhibits superior performance. We release the data and the code at https://github.com/cs-liangchen-work/NovelIE.

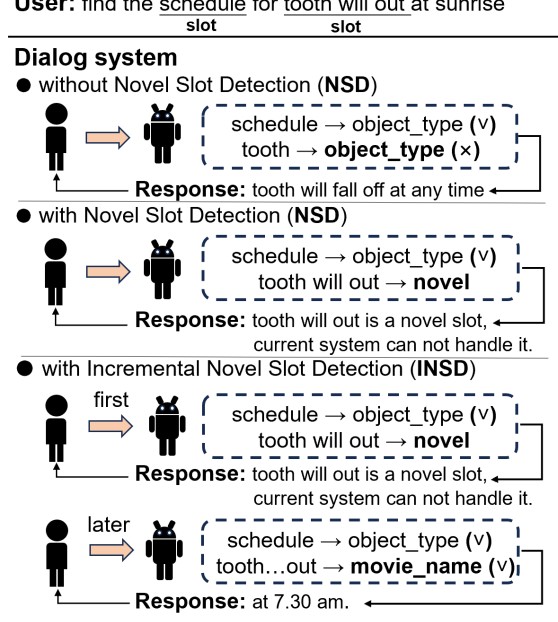

Figure 1: An example of how dialogue systems respond to the user request containing an unknown slot 'tooth will out'. Without NSD, system misclassifies slot type (the right type is 'movie name') and delivers an incorrect reply, With NSD, system returns an inability to handle the request every time. With INSD, system can provide a satisfying response in subsequent interactions.

## 1 Introduction

Slot filling (SF), a crucial component of task-oriented dialogue systems, aims to identify specific spans in user utterances (Zhang and Wang, 2016; Shah et al., 2019; Qin et al., 2021; Liang et al., 2021a; Hudeček et al., 2021; Yan et al., 2022). As dialogue systems are widely used in daily life, traditional slot filling needs to be quickly adapted to scenarios with previous unseen types (Shah et al., 2019; He et al., 2020; Liu et al., 2020b), which is known as a novel slot detection (NSD) task (Wu et al., 2021, 2022).

However, prior research in NSD only focuses on identifying new slots (Wu et al., 2021, 2022), which is too idealized and does not bring practical improvements to the dialogue system. Consider the user's request contained an unknown type of slot 'tooth will out' in Figure 1, the dialogue system with novel slot detection just provides a reply that is not wrong but less than satisfactory, and will deliver an inability to handle it all the time. Actually, dialogue systems need a specialized procedure to deal with discovered new slot or entity types.

In this paper, we focus on a more practical setting: after the model recognizes new types, the dialogue system will view them as known and can return satisfying replies to user requests concerning

---

*Work was done under the guidance of Jian Liu et al. when Liang in Beijing Jiaotong University, Li and Guan provided some experimental results.

†Corresponding author.

these types in subsequent interactions, as shown at the bottom of Figure 1. We start with a collection of annotated examples with some pre-defined types. Our goal is to identify examples with novel types (i.e., types not contained in the pre-defined set), and then, the new types are seen as known, and we should train model to have the ability to identify entities of these types, therefore resulting in incremental learning (Cao et al., 2020; Monaikul et al., 2021; Yu et al., 2021).

We separate the dialogue system to deal with novel types as two major phrases: 1) model discovers slots with unknown classes, 2) training model to possess the capability to handle new classes. We call this unified process incremental novel slot detection (INSD). For identifying new slots, we employ the set prediction method to generate a great number of triples $(start, end, type)$, and subdivide the *non-predefined entity* triple into the *non-entity* triple and the *novel-entity* triple to extract novel spans. Given that the NSD task lacks a clear objective for fitting due to unseen information of new classes at the training stage, we further provide two strategies to make the model better suit this task, which can improve robustness and encourage model to generate diverse outputs. For the INSD task, we construct two datasets based on slot filling dataset (Coucke et al., 2018; Hemphill et al., 1990). It is generally challenging to mitigate the catastrophic forgetting (McCloskey and Cohen, 1989) during the process of INSD, inspired by previous works (Cao et al., 2020; Monaikul et al., 2021; Liu et al., 2022; Xia et al., 2022), we propose a query-induced strategy to realize explicitly knowledge learning from stored data and implicitly knowledge transferring from old model together.

We evaluate the proposed method on two common benchmarks and two datasets we constructed. From the results, our approach exhibits promising results on INSD and NSD (§ 5). We also perform ablation studies to assess our method of discovering new classes and solving catastrophic forgetting (§ 6.1). Finally, we conduct further experiments to explore the NSD model's ability to identify in-domain and novel slots (§ 6.2).

To summarize, the contributions of our work are three-fold:

- We define incremental novel slot detection (INSD) aligned with real-world scenarios to improve the dialogue system's ability to deal with new slot types, and we provide effective strategies to overcome catastrophic forgetting during incremental learning.

- We propose a novel slot detection (NSD) model with set prediction, which demonstrates broader applicability.

- Results show that our method delivers the best performance on INSD and NSD tasks.

## 2   Related Work

**Novel Slot Detection.**   Prior works have extensively studied slot filling (SF) task (Zhang and Wang, 2016; Bapna et al., 2017; Rastogi et al., 2017; Shah et al., 2019; Rastogi et al., 2020; Liang et al., 2021b; Yan et al., 2022) to recognize specific entities and fill into semantic slots for user utterances. Recently, Wu et al. (2021) define novel slot detection (NSD) to extract out-of-domain or potential new entities and propose a two-step pipeline to detect them, which has attracted much attention. Subsequent works focus on the end-to-end paradigm (Liang et al., 2019) to directly obtain novel entities without intermediate steps. Wu et al. (2023) design a bi-criteria active learning framework and Wu et al. (2022) introduce semi-supervised learning scheme to perform iterative clustering. In contrast to the previous method, we propose a generative method based on set prediction to produce new entity triples.

The set prediction is an end-to-end method that enables model to directly generate the final expected set, which was proposed in DETR (Carion et al., 2020) to solve object detection task. Researchers have introduced it to natural language processing (NLP). Tan et al. (2021) propose a sequence-to-set network to handle the complexity of nested relationships in named entity recognition task. Dongjie and Huang (2022) obtain multimodal inputs representation by DETR's decoder.

**Incremental Learning.**   Incremental learning (Wang et al., 2019; Madotto et al., 2020; Kim et al., 2023) has been introduced to simulate the setting of type/class-increasing in real-world scenarios, which is mainly researched in information extraction (IE) (Wei et al., 2022; Zhang and Chen, 2023). Different from assuming data arrived with definite entity types at each stage (Cao et al., 2020; Monaikul et al., 2021), we define a new task *incremental novel slot detection* that type is unknown and needs to discover in the new data stream.

Catastrophic forgetting (McCloskey and Cohen, 1989) is a long-standing problem in incremental learning. Current approaches to overcome this issue can be roughly classified into two groups. The first is knowledge distillation (KD) (Li and Hoiem, 2017; Monaikul et al., 2021; Yu et al., 2021; Zheng et al., 2022; Kang et al., 2022), which transfers the knowledge from the previous model into the current model. The second is data retrospection (Cao et al., 2020; Liu et al., 2022; Xia et al., 2022), which samples old data and stores them in a memory buffer with a fixed capacity. In this paper, we propose a query-induced strategy to enhance KD and data retrospection.

## 3 Approach

### 3.1 Problem Definition

Novel slot detection (NSD) is a fresh task in slot filling (SF). For a given sentence $S$, a NSD model aims at extracting potential new entities or slots in $S$ (Wu et al., 2021). The slot in the NSD dataset $D$ consists of two type sets: pre-defined type set $T_p$ and novel type set $T_n$. Sentences in the training set only contain $T_p$, whereas sentences in the test set include both $T_p$ and $T_n$.

In realistic scenarios, once the model has discovered new slot classes, these types should be treated as known. Therefore, we define a new task *incremental novel slot detection* (INSD) aligned with real-world scenarios. Following previous work in incremental learning (Cao et al., 2020; Monaikul et al., 2021), we adopt the assumption that the entity class arrived at different time points. But different from them, novel types will be identified at each stage and be seen as known in the next stage in our setting.

For INSD task, we construct the dataset $D = \{D_1, D_2, ..., D_k\}$ with $k$ stage, where $D_1$ is annotated and others is unlabeled. Each $D_i$ in $D$ contains a slot type set $T_{pi} = \{t_1, t_2, ...\}$, where types in $T_{p1}$ are pre-arranged in the initial stage and $T_{pi}(i > 1)$ is composed of new types detected by model $M_{i-1}$ ($M_{i-1}$ is training at $i-1$ stage). In other words, the slot classes that are predicted as novel in $i-1$ stage will be viewed as in-domain types in $i$-th stage. At step $i$, we first apply the model $M_{i-1}$ to annotate $D_i$ with slot types $T_{pi}$, which are new discovered classes at step $i-1$. Then, we train $M_{i-1}$ on labeled $D_i$ and get $M_i$. $M_i$ also has the competence to identify novel classes contained in next-stage data. The process is shown at the top of Figure 2.

### 3.2 Novel Detection with Set Predicted Network

Next, we describe the proposed NSD model, which is depicted at the bottom-left of Figure 2. We first present the details of the model structure, which consists of two main components: *query representation* and *learning and bipartite matching*. Finally, we show how to extract novel entities.

**Query Representation.** Our model locates novel entities by recognizing the start positions and end positions based on a group of queries. Considering the information of the novel entity is entirely invisible in NSD task, we employ a random initialization strategy for the representation matrix of queries and optimize them by the model itself during training, which is distinct from the joint encoding approach in the machine reading comprehension (MRC) model (Liu et al., 2020a). Let $Q_s \in \mathbb{R}^{d \times l}$ and $Q_e \in \mathbb{R}^{d \times l}$ stand for the embedding matrix of queries $Q = \{q_1, .., q_l\}$ to predict start position and end position respectively, where $l$ is the number of queries and $d$ is their dimension.

**Learning and Bipartite Matching.** For the given sentence $S$, we adopt BERT-large (Devlin et al., 2018) to encode it and get representation $H$. By fusing $Q_s$, $Q_e$ and $H$, the model can generate triples $(s.e, t)$, which is composed of the start position $s$, end position $e$ and slot type $t$. Specifically, we first calculate probability distributions of each token being start index and end index, as follows:

$$P_s = softmax(Q_s \cdot H) \qquad (1)$$

$$P_e = softmax(Q_e \cdot H) \qquad (2)$$

Then we integrate the above positional probability information into $H$ to get span representation, which can be regarded as an attention mechanism:

$$H_{span} = \sum_i ((P_s^{(i)} \cdot H^{(i)}) + (P_e^{(i)} \cdot H^{(i)})) \quad (3)$$

where the superscript $^{(i)}$ means the $i$-th value in matrix. Finally, we predict the slot class of $H_{span}$, which comprises pre-defined slot types and label 'O' (indicates that a span does not belong to in-domain slot types). We compute the probability distribution of each class as follows:

$$P_t = T \cdot H_{span} + b \in \mathbb{R}^{1 \times (j+1)} \qquad (4)$$

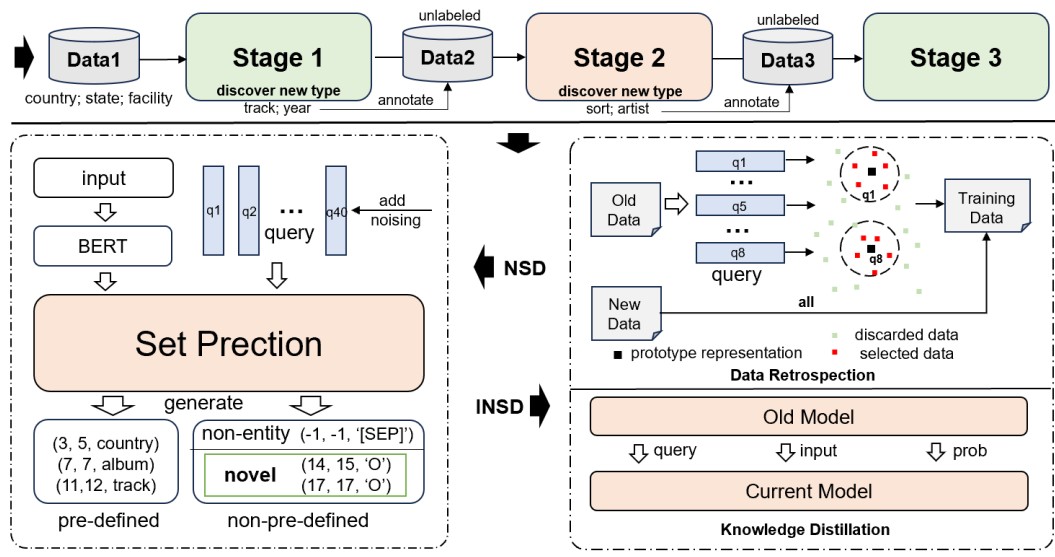

Figure 2: The overview of our method. At each stage, NSD model can discover novel slot types and we annotate next-stage data with these types. Our proposed NSD model is illustrated at the bottom left and the approach of overcoming catastrophic forgetting is shown at the bottom right.

where $T$ and $b$ are parameters in the linear layer, and $j$ is the number of pre-defined entity type.

Given that a sentence may contain one or more entities, we use hungarian algorithm (Kuhn, 1955) to optimize the matching of triples between the predicted entity $y = (s.e, t)$ and golden entity $\hat{y} = (\hat{s}, \hat{e}, \hat{t})$.

For training model, we adopt the following loss:

$$\mathcal{L}(y, \hat{y}) = -\log(w_s P_s(s|\hat{s}) + w_e P_e(e|\hat{e}) + w_t P_t(t|\hat{t})) \quad (5)$$

where $w_s$, $w_e$ and $w_t$ are the weights for different tokens and entity types.

**Novel Type Prediction.** We modify the model's loss to predict novel entities. Specifically, the value in $w_b$ and $w_e$ for the last token '[SEP]' and the value in $w_t$ for the label 'O' are both set to a smaller value than others. After training the model on the corrected loss, there are two forms triplet for non-pre-defined entities: i) $(s_{sep}, e_{sep}, t)$: the span is the last token '[SEP]' with any label, ii) $(s, e, O)$: the label 'O' with any span. By assigning appropriate values for $w_b$, $w_e$ and $w_t$, we can treat the second form $(s, e, O)$ as novel entity triple.

We further present two strategies to make the model better suit the NSD: i) *contrastive learning*: we pull the entity embeddings of the same type together in a mini-batch (Zhang et al., 2021) to improve model robustness. ii) *noise-induced adversarial learning*: we propose adversarial training with noise injection to alleviate the overfitting

of the model to in-domain types and promote the model to generate diverse triples. Random values are added to the query representation $Q_s$ and $Q_e$ at the training stage.

### 3.3 Query Enhanced Incremental Learning

Based on constructed NSD model, we now introduce our approach to mitigate the problem of catastrophic forgetting during the process of INSD, separated by data retrospection and knowledge distillation with query representation enhancement. We summarize the procedure in Algorithm 1.

**Query Enhanced Data Retrospection** Data retrospection method manages a constrained memory buffer $\mathcal{B}$ with a capacity of $c$ to store a small number of old examples. The capacity size $c$ is constant due to data storage costs. When new data comes, we will update the memory buffer to release allocated space for accommodating new types. Therefore, selecting appropriate examples from old data is crucial for data retrospection. Prototype representation has been successfully utilized in samples filtering (Cao et al., 2020; Zalmout and Li, 2022), which is a center vector integrating all features from given dataset. The calculation is as follows:

$$H_c = \frac{1}{N} \sum_{n=1}^{N} H_n \quad (6)$$

where $N$ is the number of data, $H_n$ is the embedding vector of the first token '[CLS]' in a sentence and is seen as the sentence representation.

We believe there are distinctions between slot classes, attributed to differences in data volume and learning difficulty. Consequently, we propose a new prototype learning with a query-induced data split strategy, rather than uniformly partitioning the space according to slot types. We first leverage query to cluster data: the sentences in which novel entities were successfully identified by query $q_i$ are grouped together as a subset. Then we compute the prototype representation of the subset and calculate the distance between the center and each sentence in subsets. The sentence with a small distance will be sampled and stored in the buffer $\mathcal{B}$.

In our method, we treat data at each stage equally, so the memory $\mathcal{B}$ is divided evenly according to the number of stages. At stage $i$, we discard $c/i$ data in $\mathcal{B}$ and utilize that memory buffer to store the examples sampled in $D_i$.

**Query Enhanced Knowledge Distillation**
Knowledge distillation addresses catastrophic forgetting by transferring the knowledge already acquired in the previous model. At $i$-th stage, $M_{i-1}$ acts as a teacher model that possesses the ability to identify old slot classes, and $M_i$ serves as a student model. By distilling the knowledge from $M_{i-1}$ to $M_i$, $M_i$ can largely overcome the catastrophic forgetting for old types.

The distillation module of previous methods generally contains encoded sentence representation and predicted probability distribution (the probability of the start position, end position, and type classification in our model). The loss is calculated as follows:

$$\mathcal{L}_{sent} = \text{MSE}([H]_{i-1}, [H]_i) \qquad (7)$$

$$\mathcal{L}_p = \text{KL}([P]_{i-1}, [[P]_i), P \in \{P_s, P_e, P_t\} \quad (8)$$

where KL represents the KL divergence loss and MSE denotes the mean squared error function. $[\cdot]_{i-1}$ and $[\cdot]_i$ are the value computed by model $M_{i-1}$ and $M_i$, respectively. Considering that the query plays a crucial role in the extraction of span and corresponding type, we also distill the query representation:

$$\mathcal{L}_{query} = \text{MSE}([q]_{i-1}, [q]_i) \qquad (9)$$

Finally, we train the model with the following loss:

$$\mathcal{L}_{kd} = \mathcal{L}_{sent} + \mathcal{L}_p + \mathcal{L}_{query} \qquad (10)$$

**Algorithm 1** Addresses Catastrophic Forgetting

**Input**: training set $D = \{D_1, D_2, ..., D_k\}$ with $k$ stage, memory buffer $\mathcal{B}$ with capacity $c$, NSD model $M$

1: **for** $i = 1,2,...,k$ **do**
2:      get training data $D_i \cup \mathcal{B}$
3:      train model $M_i$ and distill from $M_{i-1}$
4:      split $D_i$ based on query and select $c/i$ data
5:      release $c/i$ space and store selected data
6: **end for**

| Dataset | Train | Dev. | Test | Slot-Type |
|---------|-------|------|------|-----------|
| Snips   | 13,084 | 700 | 700 | 39 |
| ATIS    | 4,478 | 500 | 893 | 44 |

Table 1: Statistics of Snips and ATIS.

**Query Enhancement Integration** We integrate the aforementioned two query enhancement strategies to realize explicitly knowledge learning from old data and implicitly knowledge transferring from previous model together. We summarize the procedure in Algorithm 1.

## 4 Experimental setups

**Experimental Setting.** We conduct experiments on two slot filling datasets: Snips (Coucke et al., 2018) and ATIS (Hemphill et al., 1990). Snips is annotated with 13,084 sentences and has 39 slot types. ATIS provides 4,478 training examples with 44 slot types. The statistics of the datasets are shown in Table 1.

For NSD task, we follow the data setup in Wu et al. (2021). We employ the *remove* processing strategy that randomly selects several classes as unknown and the sentences containing unknown slots are deleted from the training set. For metrics, we apply the span-level precision (P), recall (R), and f1-scores (F1).

For INSD task, we construct two multi-stage datasets based on Snips and ATIS. We divide the official training set into $k$ subsets $\{D_1, D_2, ..., D_k\}$ and assign slot types $\{T_{p1}, T_{p2}, ..., T_{pk}\}$ for each stage. We process the $D_i$ as follows: 1) directly remove the sentences including types in $T_{p\cdot}$ ($\cdot > i$), 2) label the slot values belonging to $T_{p\cdot}$ ($\cdot < i$) with 'O', and the corresponding text tokens are replaced with 'MASK', 3) $D_i$ ($i > 1$) is unannotated and needs NSD model to discover and label novel slots. We exploit four metrics to assess model perfor-

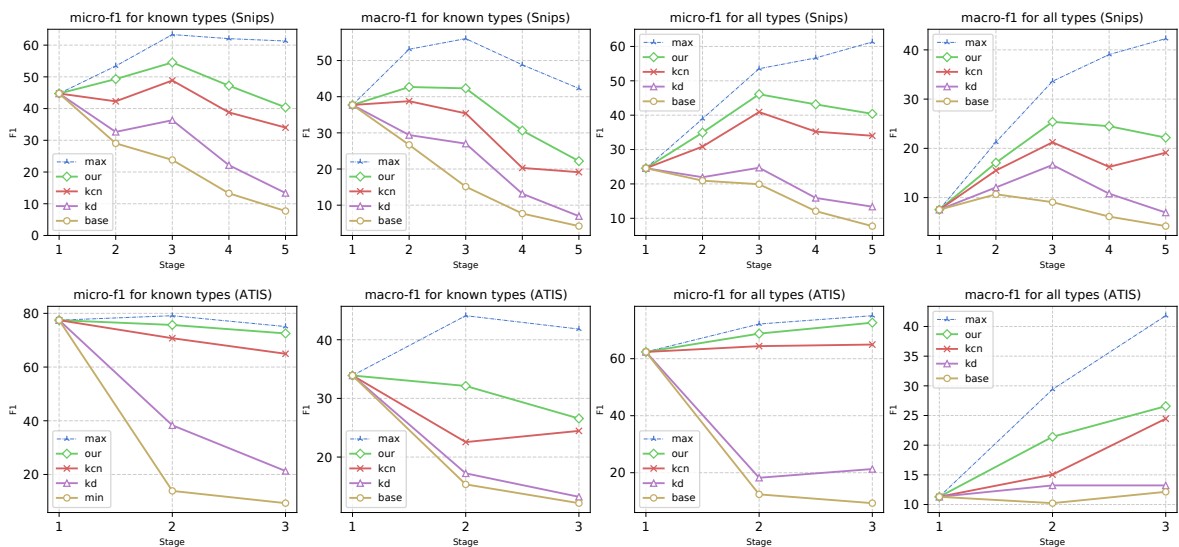

Figure 3: The results of the INSD model at each stage on the test set of Snips (first row) and ATIS (second row) with four evaluation metrics. 'max' refers to the upper bound, which trains model with all available data. 'base' means the lower bound without incremental learning strategy. For $i$-th stage, 'known types' is the types in stage $1 \sim i$.

| Dataset | Model | 5% | | 10% | | 15% | |
| --- | --- | --- | --- | --- | --- | --- | --- |
| | | IND | NSD | IND | NSD | IND | NSD |
| Snips | GDA-min (Wu et al., 2021) | 93.14 | 29.73 | 85.28 | 14.28 | 90.07 | 31.96 |
| | GDA-different (Wu et al., 2021) | 93.10 | 31.67 | 86.22 | 15.06 | 90.18 | 32.19 |
| | Our method | 93.57 | 39.49 | 91.52 | 35.19 | 91.30 | 40.72 |
| ATIS | GDA-min (Wu et al., 2021) | 90.14 | 10.17 | 93.57 | 23.18 | 93.92 | 50.92 |
| | GDA-different (Wu et al., 2021) | 90.68 | 10.27 | 94.01 | 22.98 | 93.88 | 43.78 |
| | Our method | 94.84 | 39.20 | 93.04 | 38.53 | 93.37 | 61.71 |

Table 2: Model performance (f1-score) on two benchmarks. 'IND' and 'NSD' denote pre-defined and novel entity types respectively. '·%' is the proportion of potential new classes.

mance: micro-averaged f1 and macro-averaged f1 for current known types, and micro-averaged f1 and macro-averaged f1 for all types.

**Implementations.** We employ BERT (Devlin et al., 2018) as contextualized encoder to implement our model. For constructed multi-stage INSD dataset, the k is set to 5 in Snips and 3 in ATIS. The model is trained for 30 epochs with a learning rate of 1e-5 and a batch size of 8. The query number is selected in {20, 40, 60, 80} and the weight to generate novel slot span is chosen from {1e-1, 1e-3, 1e-5}. We set the capacity of memory as 100.

**Baselines.** For INSD task, we compare our approach to the knowledge distillation model **KL** (Monaikul et al., 2021) and the data retrospection model **KCN** (Cao et al., 2020). For NSD task, we compare the proposed method to the pipeline model GDA (Wu et al., 2021) with two distance

strategies **GDA-min** and **GDA-different**.

## 5 Overall Results

In this section, we present the main results of incremental novel slot detection (INSD) framework and novel slot detection (NSD) model.

**Model Performance of INSD** We visualize the results of our proposed approach and baselines during the incremental learning process in Figure 3. It can be observed that our method demonstrates the best performance compared with baseline approaches for all metrics and datasets, achieving 22.73% macro-f1 and 41.75% micro-f1 on Snips and 26.37% macro-f1 and 72.53% micro-f1 on ATIS respectively. Such results indicate that our method shows the best ability to overcome catastrophic forgetting for the INSD task. From the figure, we also see that the data retrospection method

(KCN) brings much higher f1 improvements than the knowledge distillation approach (KL), which suggests that the explicit data-storing strategy is more effective. Our proposed approach introduces queries to enhance them and can further improve model performance.

**Model Performance of NSD** Table 2 presents the performance of our method as well as baselines on two datasets. For extracting novel slots (NSD), we can see that our model outperforms previous methods by a large margin for all datasets and scenarios. For example, our method beats GDA-different by gaining improvements of 7.82%, 20.13%, and 8.52% in terms of F1-scores in different settings on Snips. For identifying in-domain slot (IND), our model also obtains comparable results compared with baselines, but it exhibits inferior performance in some settings such as 15% sampling on ATIS. By comprehensively analyzing, we can find that our proposed model demonstrates better capability in discovering novel types of slots than known slots.

# 6 Discussion

We conduct further experiments to investigate the proposed INSD framework and NSD model, separated by ablation study and the learning ability of the NSD model.

## 6.1 Ablation Experiment

**Ablation Study of INSD** We perform ablation experiments to investigate the proposed INSD method. The results of 3-th and 5-th stage are depicted in table 3. As can be seen, our query enhancement approach can lead to better performance, attaining 1% and 6% increases in terms of F1 separately in knowledge distillation and data storage. Results demonstrate that query representation is important in novel slot extraction and indicate that first clustering the data via queries and then calculating the prototype representation can select beneficial samples.

**Ablation Study of NSD** We conduct an ablation study to explore the impact of contractive learning (CL) and noise-induced adversarial learning (NL) on our NSD model. The results are presented in Table 4. As expected, CL and NL can both yield greater F1 scores than the base model. Overall, the NL method is more effective because noise can contribute to generating diverse outputs. In addi-

| Setting | Stage 3 | | | Stage 5 | | |
|---|---|---|---|---|---|---|
| | P | R | F1 | P | R | F1 |
| Full model | 58.30 | 51.53 | 54.71 | 54.41 | 33.87 | 41.75 |
| Base model | 26.74 | 21.46 | 23.81 | 11.38 | 5.83 | 7.71 |
| w KL | 42.04 | 31.99 | 36.33 | 37.18 | 8.16 | 13.38 |
| w KL(Q) | 40.67 | 33.69 | 36.85 | 45.77 | 8.51 | 14.36 |
| w DataStore | 54.12 | 44.51 | 48.85 | 44.96 | 27.34 | 34.01 |
| w DataStore(Q) | 56.57 | 52.64 | 54.53 | 51.34 | 33.29 | 40.39 |

Table 3: Ablation study of INDS model on Snips dataset. We report the micro-averaged P, R, and F1-score for current known types at the 3-th and 5-th stage. 'Base' means the model without incremental learning strategy. '(Q)' refers to a method with query enhancement.

| Dataset | Setting | 5% | 10% | 15% | $\Delta$avg |
|---|---|---|---|---|---|
| Snips | Base model | 34.25 | 31.16 | 33.55 | |
| | w CL | 35.24 | 29.23 | 35.79 | 0.4 ↑ |
| | w NL | 36.63 | 33.16 | 39.41 | 3.4 ↑ |
| | Full model | 39.49 | 35.19 | 40.72 | 5.5 ↑ |
| ATIS | Base model | 37.85 | 35.93 | 58.20 | |
| | w CL | 38.11 | 35.19 | 59.28 | 0.2 ↑ |
| | w NL | 38.38 | 35.14 | 59.82 | 0.4 ↑ |
| | Full model | 39.20 | 38.53 | 61.71 | 2.5 ↑ |

Table 4: Ablation study of our NDS model on two datasets. 'Base model' stands for the base set prediction model without 'CL' (contrastive learning) and 'NL' (noise-induced adversarial learning).

tion, unlike typical NLP tasks, NSD task lacks a clear objective for fitting and is trained with slot filling (SF) task, which makes it essential to introduce noise during training stage. By employing both CL and NL, the model demonstrates a much higher improvement in F1-score, surpassing the sum of the gains obtained by each method individually. A reasonable explanation is that our proposed CL can enhance model robustness by encouraging entity embeddings of the same type to be close, which mitigates the negative impact of noise when training model.

**The Impact of Weight Parameters and Query Number** We study the effect of weight and query number on model performance. For simplicity, we set fixed entity type weight $w_t$ for the label 'O' and assign the same value for the last position weight $w_s$ and $w_e$. Figure 4 depicts the results. We can conclude that: 1) Our model exhibits high sensitivity to the weight and query number. 2) As the count of novel entities increases (5%→10%→15%), the

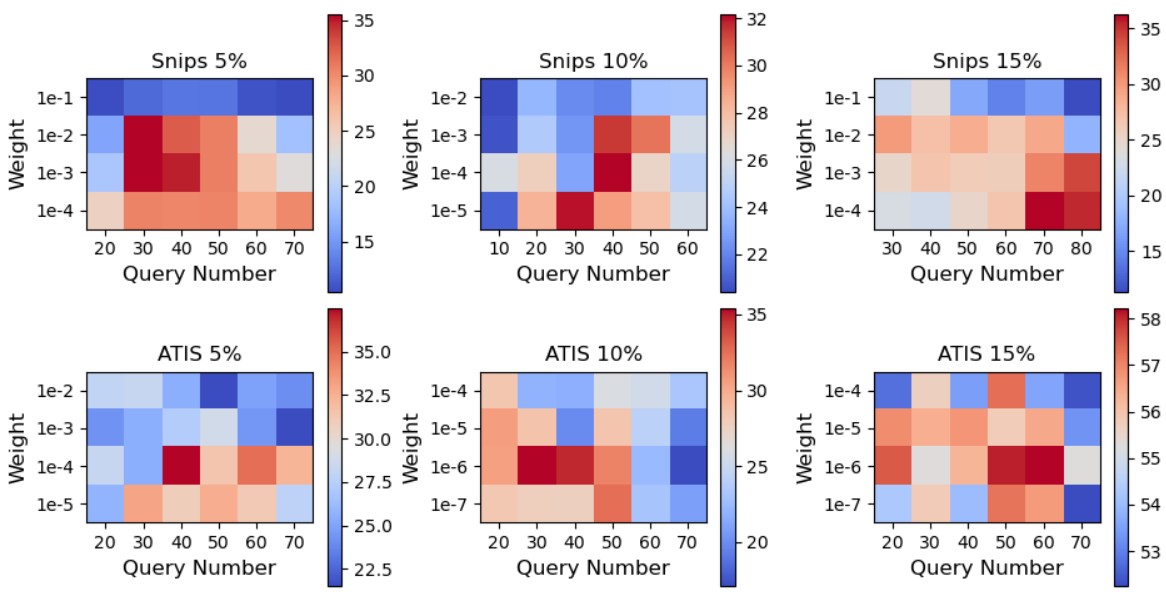

Figure 4: The impact of the count of query and the value of weight.

model requires a greater number of queries and reduced weight values, which both encourage our model to generate a larger quantity of novel entity triplets $(s, e, t)$. Specifically, the increasing number of queries allows the model to learn various query representations for different novel slot types, while decreasing the weight values can result in a higher count of $(s, e, O)$ triplets (the label 'O' with any span) and a lower count of $(s_{sep}, e_{sep}, t)$ triplets (the span is the last token '[SEP]' with any label).

## 6.2 Learning Ability of NSD Model

**The Ability to Discover Novel Entity** We want to explore the NSD model's inclination in identifying novel slot classes, are newly discovered and pre-defined types similar or dissimilar? To answer this, we first randomly select a subset of classes as in-domain and then choose novel classes from the remains based on label semantics similarity and entity representation similarity. We conduct experiments in two settings 'Sim' and 'Disim'. The results are shown in Table 5. We can observe that the previous model GDA tends to discover novel types that differ from pre-defined types, whereas our model delivers superior performance in all settings. Results reveal the broader applicability of our model.

**Learning Ability of Queries** We investigate the ability of queries to identify various types of spans, containing non-entity, novel entity, and in-domain entity. The proportion of new and pre-defined entities extracted by each query is plotted in Figure 5,

| Model | SimL | SimM | DisimL | DisimM |
|---|---|---|---|---|
| GDA-min | 5.70 | 16.41 | 15.38 | 20.50 |
| GDA-diff | 5.71 | 16.38 | 15.41 | 20.63 |
| Our Method | 20.52 | 21.58 | 16.76 | 24.11 |

Table 5: F1 scores of different novel slot class setup. 'Sim' and 'Disim' represent whether the novel types are similar to pre-defined types. 'L' (little) and 'M' (many) stand for the number of novel types.

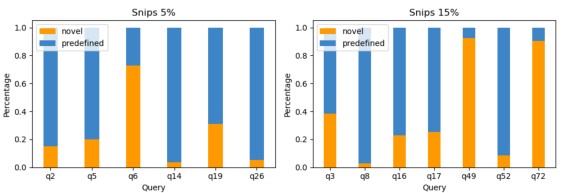

Figure 5: Visualization of the proportion of new and pre-defined slots identified by each query.

where the queries that only recognize the non-entity span were not drawn and we only present the result in 5% and 15% scenarios on Snips. From the figure, we can see that the count of queries for non-entity greatly outnumbers the queries for the entity, such as 7 entity queries and 73 non-entity queries in the 15% sampling setting, primarily due to the higher number of non-entity spans. Interestingly, it can be observed that each entity query possesses the capability to recognize both potential new and pre-defined entities in our model.

## 7 Conclusion

In this paper, we introduce a new task incremental novel slot detection (INSD) and establish a benchmark with two datasets and four metrics. Moreover, we propose a query-induced strategy to alleviate the problem of catastrophic forgetting during incremental learning and provide an effective model to identify novel slots. Our proposed approach achieves state-of-the-art performance in all datasets and scenarios.

## 8 Limitations

Although the proposed method exhibits superior performance in overcoming catastrophic forgetting and identifying novel slots, the extracted entity quantity decreases as the stages advance, and several classes cannot even be recognized. However, for incremental novel slot detection task, it is crucial to discover as many unseen classes as possible. Another limitation is that our model to detect new slots shows high sensitivity to the weight and query number, which means that researchers need to spend time to carefully tune the parameters for different datasets.

## 9 Acknowledgements

This work was supported by the Fundamental Research Funds for the Central Universities 2023JBMC058. It was also supported by the National Natural Science Foundation of China (No.62106016, 61976015, 61976016, 61876198 and 61370130), and the Open Projects Program of the State Key Laboratory of Multimodal Artificial Intelligence Systems.

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
