# OpenReview forum: "Novel Slot Detection With an Incremental Setting"
_EMNLP/2023/Conference — EMNLP 2023 Findings_

### Official Review · Reviewer_Mmoy · 2023-08-03

**Typos Grammar Style And Presentation Improvements:** 1.  The contribution of the paper is …
**Soundness:** 3

**Excitement:**

3: Ambivalent: It has merits (e.g., it reports state-of-the-art results, the idea is nice), but there are key weaknesses (e.g., it describes incremental work), and it can significantly benefit from another round of revision. However, I won't object to accepting it if my co-reviewers champion it.

**Missing References:**

Open-vocabulary slot-filling:

1. Rastogi A, Zang X, Sunkara S, et al. Towards scalable multi-domain conversational agents: The schema-guided dialogue dataset[C]//Proceedings of the AAAI conference on artificial intelligence. 2020, 34(05): 8689-8696.

2. Rastogi A, Hakkani-Tür D, Heck L. Scalable multi-domain dialogue state tracking[C]//2017 IEEE Automatic Speech Recognition and Understanding Workshop (ASRU). IEEE, 2017: 561-568.

3. Bapna A, Tur G, Hakkani-Tur D, et al. Towards zero-shot frame semantic parsing for domain scaling[J]. arXiv preprint arXiv:1707.02363, 2017.

4. Dai Y, Zhang Y, Liu H, et al. Elastic CRFs for Open-Ontology Slot Filling[J]. Applied Sciences, 2021, 11(22): 10675.


Incremental dialog learning

1. Wang W, Zhang J, Li Q, et al. Incremental learning from scratch for task-oriented dialogue systems[J]. arXiv preprint arXiv:1906.04991, 2019.

2. Madotto A, Lin Z, Zhou Z, et al. Continual learning in task-oriented dialogue systems[J]. arXiv preprint arXiv:2012.15504, 2020.

3. Kim G, Xiao C, Konishi T, et al. Open-World Continual Learning: Unifying Novelty Detection and Continual Learning[J]. arXiv preprint arXiv:2304.10038, 2023.


**Paper Topic And Main Contributions:**

This paper proposes a task called incremental novel slot detection that continually detects new slots, labels new slots, and retrains the slot detection model for the next-round detection. The authors adapt the SNIPS and ATIS, two classical slot-filing datasets to fulfill this setting. They proposed to combine contrastive learning and noise-induced adversarial learning for novel type prediction and propose a Query Enhanced Knowledge Distillation to alleviate the catastrophe-forgetting problem during incremental learning.  Experimental results show the efficacy of the proposed framework when compared with a novel-type prediction baseline and two continual learning baselines.

**Questions For The Authors:**

1. Table 2. For the Snips dataset, why the NSD results in a decrease from 5% to 10% and an increase from 10% to 15%?

2. Line 374. Step 2, Why replace the text token with MASK, and slot values belonging (T_p<i) are labeled with O? Should it be T_p>i?

3. Line 190 After a new slot value is detected at the current stage, the novel slots will be viewed as in-domain types. Does that mean only the set of the new slot names becomes available? Or all the labeled data for those new slots are available?


**Reasons To Accept:**

1. Propose a research problem that is more relevant to the realistic setting, i.e., incremental slot detection (both out-of-distribution and in-distribution), which requires continually detecting new slot values and deploying the new slot prediction models. This could be an interesting research topic for online dialog system development.

2. For slot detection, the paper utilizes learnable query vectors for feature matching and sequence prediction and use Hungarian algorithm to optimize the labeled triples. the same query can also be used to retrieve some representative data from the training set for incremental learning to avoid the catastrophic forgetting problem. Knowledge distillation is also used for fast model adaptation. This framework could potentially be useful and efficient in real dialog systems.

**Reasons To Reject:**

1. Lacking strong baselines. The incremental learning for dialog systems is a classical topic, many different approaches, like architecture-based, memory-based, and data-retrieval-based are proposed in previous work[1], and also compared with strong multi-task learning baseline.  Since this paper focuses on incremental learning, it needs more comparison with different baselines.

2. Lacking more complex benchmarks to verify the effectiveness of the method. Current benchmarks are small and toylike. More realistic dialog datasets like MultiWOZ, SGD, and dataset on DialogStudio should be considered.  In addition, ChatGPT or GPT-4 results on this benchmark should be included for more comparison. This may not be required, but it's crucial to demonstrate how hard and challenging the new task is.

[1] Madotto A, Lin Z, Zhou Z, et al. Continual learning in task-oriented dialogue systems[J]. arXiv preprint arXiv:2012.15504, 2020.


**Reproducibility:**

4: Could mostly reproduce the results, but there may be some variation because of sample variance or minor variations in their interpretation of the protocol or method.

**Reviewer Confidence:**

4: Quite sure. I tried to check the important points carefully. It's unlikely, though conceivable, that I missed something that should affect my ratings.

---

> ### Author Rebuttal · Authors · 2023-08-29
>
> We thank you for the insightful and detailed reviews.
>
> Q1: For the Snips dataset, why the NSD results in a decrease from 5% to 10% and an increase from 10% to 15% in Table 2?
>
> A1: That is because it contains novel types that are relatively difficult to identify in 10% setting, such as ‘compartment’. In all three scenarios, the selection of novel types is random.
>
> Q2: Line 374. Step 2. Why replace the text token with MASK, and slot values belonging ($T_p<i$) are labeled with O? Should it be $T_p$>I?
>
> A2: Slot values that belong to ($T_p$<i) are labeled with O, not $T_p$>i. In our setting, we assume that the data $D_i$ at the i-th stage only includes the slot types of the current stage. For sentences containing slot types from stage j (where j>i), as shown in line 371, we directly remove these sentences from $D_i$. For sentences containing types from stages j (where j<i), we replace the corresponding text tokens with MASK and slot values with 'O' in $D_i$, which ensures that $D_i$ does not contain the slot types belonging to stages 1 to i-1.
>
> Q3: Line 190. After a new slot value is detected at the current stage, the novel slots will be viewed as in-domain types. Does that mean only the set of the new slot names becomes available? Or all the labeled data for those new slots are available?
>
> A3:  The labeled data for those new slots are available. As described in line 192, we will employ a NSD model to annotate these novel slot types within the data.
>
> Additionally, we appreciate the provided missing references and presentation adjustments. We have revised them in the final version. Thanks.

---

### Official Review · Reviewer_Eu35 · 2023-08-06

**Soundness:** 3

**Excitement:**

3: Ambivalent: It has merits (e.g., it reports state-of-the-art results, the idea is nice), but there are key weaknesses (e.g., it describes incremental work), and it can significantly benefit from another round of revision. However, I won't object to accepting it if my co-reviewers champion it.

**Paper Topic And Main Contributions:**

This paper aims to solve the problem of novel slot detection, which can discover potential new types in dialogue systems. The authors define incremental novel slot detection (INSD), which separates the dialogue system to deal with novel types as two major phrases: 1) model discovers unknown slots, 2) training model to possess the capability to handle new classes. They construct two INSD datasets to evaluate the proposed method and experimental results show that their approach exhibits superior performance.

**Questions For The Authors:**

* What's the meaning of '$j$' in Equation (4)? What is T? What does $H_{span}$ refer to?
* What is $P_s(s|sˆ)$ in Equation (5)?
* Why does pulling the entity embeddings of the same type together in a mini-batch help?

**Reasons To Accept:**

* The proposed method can improve performance of novel slot detection (NSD) significantly.

**Reasons To Reject:**

* The authors claim that their method can handle new classes with a satisfying response in subsequent interactions. However, I didn't find how a novel class bind to a concrete slot (e.g., movie_name) in databases. Is it human in the loop?
* [Line 178-179] The assumption that the entity class arrived at different time points may not be right for a real application with many users.
* The authors apply the model $M_{i−1}$ to annotate $D_i$ with slot types $T_{p_i}$, while it may contain annotation errors. How to avoid that?
* For real application, how to name $T_{p_i}$ at different state?
* I think the authors may overstate the contributions of this work. I would say the proposed method is a new novel slot detection method incorporating with incremental learning. It does not show how to handle new classes to provide a satisfying response.

**Reproducibility:**

3: Could reproduce the results with some difficulty. The settings of parameters are underspecified or subjectively determined; the training/evaluation data are not widely available.

**Reviewer Confidence:**

4: Quite sure. I tried to check the important points carefully. It's unlikely, though conceivable, that I missed something that should affect my ratings.

**Typos Grammar Style And Presentation Improvements:**

* 'i-1' -> '$i-1$'
* 'i-th' -> '$i$-th'

---

> ### Author Rebuttal · Authors · 2023-08-29
>
> We thank for the insightful and detailed reviews.
>
> Q1: How a novel class bind to a concrete slot (e.g., movie_name) in databases?
>
> A1: The model is capable of identifying whether slots belongs to novel classes, while the corresponding concrete types require human in the loop or the addition of extra classifiers. Because we are assuming that the novel types are completely unseen, we have not supplied a set of new types for alignment in our method.
>
>
> Q2: How to avoid annotation errors？
>
> A2: Sorry for not detailed description for this. In our setting, we will filter the false annotations based on the golden annotations to improve the quality of labeled data.
>
>
> Q3: For real application, how to name $Tp_{i}$ at different state?
>
> A3: This depends on the data from real-world scenarios. Assuming the model is currently in stage i, it will receive a substantial amount of user data $D$ that includes novel types. The slot types in $Tp_{i+1}$ are those new types in $D$ that the model identified.
>
>
> Q4: About the meaning of Equation (4) $P_{t} = T \cdot H_{span} + b \in \mathbb{R}^{1 \times (j+1)}$?
>
> A4: Eq.4 is employed to classify the type of the identified slot, where $H_{span} \in \mathbb{R}^{1 \times dim} $ is the vector representation of the slot span, $T$ and $b$ stand for the weight and bias in the linear layer respectively, $j$ is the number of pre-defined slot types and the addition of ‘+1’ is to consider label ‘O’. After calculation, we obtain the probability distribution $ P_{t} \in \mathbb{R}^{1 \times (j+1)}$.
>
> Q5: What is $P_{s}(s|\hat{s})$ in Equation (5)?
>
> A5: This is done to compute the cross-entropy loss between the predicted values $s$ and the golden values $\hat{s}$ concerning the start positions.
>
>
> Q6: Why does pulling the entity embeddings of the same type together in a mini-batch help?
>
> A6: For NSD task, it can establish a clearer boundary between pre-defined slot types and label ‘O’, thereby preventing the pre-defined types from affecting the process of finding novel slots within spans labeled as 'O'.
>
>
> Additionally, sorry for the typos and we have revised them in the final version. Thanks.

---

### Official Review · Reviewer_J6xJ · 2023-08-09

**Soundness:** 4

**Excitement:**

4: Strong: This paper deepens the understanding of some phenomenon or lowers the barriers to an existing research direction.

**Paper Topic And Main Contributions:**

This paper introduces a new task incremental novel slot detection (INSD) and proposes a query-induced strategy to alleviate the problem of catastrophic forgetting during incremental learning.

**Questions For The Authors:**

1. Have you tried to use self-labeling strategy like clustering to replace human-labeling?

**Reasons To Accept:**

1. The problem is interesting and valuable.
2. The experiments and analysis are solid and convincing.

**Reasons To Reject:**

Lacking more baselines. Maybe add several slot filling or incremental learning baselines.

**Reproducibility:**

4: Could mostly reproduce the results, but there may be some variation because of sample variance or minor variations in their interpretation of the protocol or method.

**Reviewer Confidence:**

4: Quite sure. I tried to check the important points carefully. It's unlikely, though conceivable, that I missed something that should affect my ratings.

---

> ### Author Rebuttal · Authors · 2023-08-29
>
> We thank you for the insightful and detailed reviews.
>
> Q: Have you tried to use self-labeling strategy like clustering to replace human-labeling?
>
> A: Thank you for your suggestion, we will consider self-labeling strategies in our future work. In this paper, the provided datasets are based on the processing of existing datasets.

---

### Official Review · Reviewer_XgD7 · 2023-08-11

**Typos Grammar Style And Presentation Improvements:** 028
**Soundness:** 2

**Excitement:**

3: Ambivalent: It has merits (e.g., it reports state-of-the-art results, the idea is nice), but there are key weaknesses (e.g., it describes incremental work), and it can significantly benefit from another round of revision. However, I won't object to accepting it if my co-reviewers champion it.

**Paper Topic And Main Contributions:**

This paper introduces a new slot detection approach for dialogue system and proposes a two-stage method , i.e., first finding the new slot types and then training the model to deal with these new slot classes, to incrementally to alleviate catastrophic forgetting issue and improve the model performance for novel slot discovery.

**Questions For The Authors:**

1. How catastrophic forgetting  influence on novel slots detection,  any supported , quantitative or qualitative experiments and analyses？

**Reasons To Accept:**

1. The paper introduce a new incremental new slot identification method to overcome catastrophic forgetting problem.

2. The paper adopts quite a lot of techniques to improve the tricky issue of new slot detection and forgetting issue, like query-induced strategy, knowledge distillation,  data retrospection, prototype representation,  query  representation  learning and bipartite matching, contrastive learning, noise-induced adversarial learning,  etc.


**Reasons To Reject:**

1. The problem that how catastrophic forgetting exerts a strong influence on novel slots detection keeps unclear.

2. The paper does not study on large language models, which may be the current SOTA  models  for novel slots detection and  their effective usage in dialogue context.

3. The method is little bit complex and hard to follow, e.g., how the method implement  the final effect in Figure?
The proposed  may be not easy to implement in real-world scenarios.

4. some experiments are missing , e.g.,  contrastive learning and  adversarial learning.

5. The comparing baselines  is only few while the proposed method is claimed to be SOTA model.

**Reproducibility:**

3: Could reproduce the results with some difficulty. The settings of parameters are underspecified or subjectively determined; the training/evaluation data are not widely available.

**Reviewer Confidence:**

3: Pretty sure, but there's a chance I missed something. Although I have a good feel for this area in general, I did not carefully check the paper's details, e.g., the math, experimental design, or novelty.

---

> ### Author Rebuttal · Authors · 2023-08-29
>
> We thank you for the insightful and detailed reviews.
>
> Q1: How catastrophic forgetting influence on novel slots detection?
>
> A1: The influence of catastrophic forgetting is in INSD (incremental novel slots detection),   not in NSD (novel slots detection). For INSD, at stage i, we can only access data $D_i$, but the data in previous stages (1 to i-1) is not available. Catastrophic forgetting will impact the extraction of slot types belonging to the previous stages.  The results are shown in Figure 3 (Section 5) and Table 3 (Section 6.1).
> In the Figure 3, ’base’ means the method without an incremental learning strategy, which trains the model only with the data at that stage, while ’max’ refers to the upper bound for solving catastrophic forgetting, which trains the model with all available data (data in stage 1 to i).
>
> Q2: Experiments for contrastive learning and adversarial learning are missing.
>
> A2: The related experiments are described in Section 6.1 (Ablation Study of NSD) and the results are shown in Table 4.
>
> Q3: The method is little bit complex and hard to follow.
>
> A3:  Thank you for this concern. We have included example codes in the Supplementary Materials and will make all codes available once the paper is accepted.
>
> Q4: About the study on large language models.
>
> A4: LLM possesses the capability to recognize all entities known to humans, while the NSD task focuses on identifying unseen slot types based on the understanding of known slot types. Therefore, using LLM to solve NSD may not be reasonable, unless it has not encountered some slot types during training.

---

### Meta-Review · Area_Chair_5YXi · 2023-09-19

**Recommendation:** 3

**Metareview:**

The paper focuses on a newly proposed task, incremental novel slot detection, and introduces a method combining multiple techniques to tackle this task and mitigate the catastrophic forgetting problem. All reviewers agree on the relevance and applicability of the new task to real-world dialogue system development, and the potential of the framework proposed.

Despite recognizing the merits of the paper in targeting an area of interest, the reviewers raise concerns about the paper due to the shortcomings in experiments, supporting methodology and baselining against already established models/methods.

---

### Decision · Program_Chairs · 2023-10-07

**Decision:**

Accept-Findings

**Comment:**

The paper focuses on a newly proposed task, incremental novel slot detection, and introduces a method combining multiple techniques to tackle this task and mitigate the catastrophic forgetting problem. All reviewers agree on the relevance and applicability of the new task to real-world dialogue system development, and the potential of the framework proposed.

Despite recognizing the merits of the paper in targeting an area of interest, the reviewers raise concerns about the paper due to the shortcomings in experiments, supporting methodology and baselining against already established models/methods.